# Albumin-Bound Fatty Acids Modulate Endogenous Angiotensin-Converting Enzyme (ACE) Inhibition

**DOI:** 10.3390/biomedicines14010103

**Published:** 2026-01-04

**Authors:** Enikő Edit Enyedi, Attila Ádám Szabó, Tamás Bence Pintér, Ivetta Siket Mányiné, Anna Pluhár, Csongor Váradi, Emese Bányai, Attila Tóth, Zoltán Papp, Miklós Fagyas

**Affiliations:** 1Division of Clinical Physiology, Department of Cardiology, Faculty of Medicine, University of Debrecen, 4032 Debrecen, Hungaryszabo.attila@med.unideb.hu (A.Á.S.); pinter.tamas@med.unideb.hu (T.B.P.); sivi@med.unideb.hu (I.S.M.); p.anna09@mailbox.unideb.hu (A.P.); atitoth@med.unideb.hu (A.T.); pappz@med.unideb.hu (Z.P.); 2Kálmán Laki Doctoral School, University of Debrecen, 4032 Debrecen, Hungary; 3Department of Surgery, Faculty of Medicine, University of Debrecen, 4032 Debrecen, Hungary; varadi.csongor@med.unideb.hu; 4Department of Emergency Medicine, Faculty of Medicine, University of Debrecen, 4032 Debrecen, Hungary; banyai.emese@med.unideb.hu

**Keywords:** angiotensin-converting enzyme, free fatty acid, ACE inhibition, human serum albumin, albumin–fatty acid binding, renin–angiotensin–aldosterone system, RAAS modulation, NEFA, FFA, non-esterified fatty acid, HSA

## Abstract

**Background/Objectives**: Human serum albumin (HSA) is a major endogenous inhibitor of angiotensin-converting enzyme (ACE) and helps fine-tune the activity of the renin–angiotensin–aldosterone system (RAAS), thereby potentially influencing the development of cardiovascular (CV) diseases. As the principal transport protein for free fatty acids (FFAs), HSA may have its ACE-inhibitory capacity modified by its FFA cargo and, through this mechanism, may also affect CV disease risk. We therefore tested the hypothesis that the composition of HSA-bound FFAs determines the magnitude of endogenous ACE inhibition. **Methods**: We quantified endogenous ACE inhibition and examined the effect of FFA concentration on this inhibition in clinical patients (n = 161 and n = 101, respectively). We measured the effects of HSA treated with saturated, monounsaturated, and polyunsaturated FFAs, as well as FFA-free HSA, on recombinant ACE and on tissue ACE. **Results**: Endogenous ACE inhibition was stronger in patients with higher serum HSA concentrations (Spearman’s rho = 0.422, 95% CI 0.281–0.544, *p* < 0.001), whereas total FFA concentration was not associated with endogenous ACE inhibition (Spearman’s rho = 0.088, *p* = 0.38, n = 101). However, removal of free fatty acids substantially worsened the ACE-inhibitory effect of HSA on recombinant ACE (charcoal-treated HSA: IC_50_ = 23.24 [19.40–29.78] g/L vs. control HSA: 7.84 [6.58–9.75] g/L, *p* < 0.001) and on tissue ACE isolated from lung, heart, and lymph node. FFA chain length, degree and position of unsaturation, and cis/trans configuration all differentially modulated endogenous ACE inhibition. Among saturated fatty acids, stearic acid (IC_50_ = 7.98 [7.04–9.23] g/L), and among omega-3 and omega-6 fatty acids, α-linolenic (IC_50_ = 5.60 [4.28–6.15] g/L) and γ-linolenic acids (IC_50_ = 5.09 [4.28–6.15] g/L) produced the greatest enhancement of the ACE-inhibitory capacity of HSA. **Conclusions**: The present results indicate that HSA concentration relates to endogenous ACE inhibition in serum, and in vitro experiments demonstrate that HSA-bound FFAs can modulate HSA-mediated ACE inhibition, a mechanism that may be relevant to cardiovascular physiology and disease.

## 1. Introduction

Dysregulation of the renin–angiotensin–aldosterone system (RAAS) underpins the pathogenesis of several major cardiovascular diseases [1,2]. In these high-burden conditions—such as hypertension, heart failure, and stroke—randomized trials and meta-analyses show that inhibition of angiotensin-converting enzyme (ACE) reduces cardiovascular events and mortality [3,4]. ACE, present in both membrane-bound and soluble forms, is prominently expressed on vascular endothelium and at epithelial surfaces in key organs (e.g., lung, heart, kidney, intestine) and generates angiotensin II, the principal RAAS effector [5]. Consequently, ACE inhibitors exert their benefits chiefly by limiting angiotensin II formation and downstream AT1-receptor–mediated vasoconstriction, sodium retention, and adverse remodeling [6].

Beyond pharmacological inhibition, ACE activity is also shaped in vivo by endogenous regulation. At the genetic level, ACE expression is largely influenced by the common insertion/deletion (I/D) polymorphism—DD carriers show approximately 1.5–2-fold higher circulating ACE than II carriers [7]—yet this variation has little or at most inconsistent impact on cardiovascular morbidity and mortality [8]. A plausible explanation is that human serum albumin (HSA) acts as a strong endogenous inhibitor of ACE at physiological concentrations, thereby buffering genotype-driven differences in ACE expression [9]. At physiological concentrations (~35–52 g/L), HSA suppresses circulating ACE activity by almost 90%, and it also inhibits membrane-bound ACE where local HSA concentrations within tissues are sufficiently high [9].

Nonetheless, multiple studies have confirmed that low—or even low-normal—HSA is consistently associated with higher cardiovascular morbidity and mortality [10,11]. Lower HSA may imply weaker HSA-mediated endogenous ACE inhibition; thus, ACE inhibitors may be particularly effective in such patients by pharmacologically substituting for a missing physiological brake on angiotensin II generation [12]. Notably, HSA is the predominant plasma carrier for a wide range of biomolecules and drugs, including non-esterified (free) fatty acids (FFA). HSA contains multiple, non-equivalent long-chain fatty-acid binding sites, and fatty-acid binding induces reproducible conformational and surface-property changes that can remodel HSA’s interactions with partner proteins [13,14,15].

FFAs themselves may relate to cardiovascular risk. Higher intake of the plant omega-3 fatty acid α-linolenic acid (ALA) is associated with modest reductions in cardiovascular diseases (CVD) and coronary mortality, and ALA-rich flaxseed lowers blood pressure in randomized trials [16,17]. Likewise, greater intake and circulating levels of the omega-6 fatty acid linoleic acid are associated with lower risks of CVD and mortality [18,19]. However, elevations in total circulating FFAs have been unfavorably linked to cardiovascular risk [20], even as specific FFA species may confer benefits. These observations raise the questions central to this work: do the level of HSA-bound FFAs modulate the degree of endogenous ACE inhibition, and can specific FFAs confer cardiovascular benefit, at least in part, by enhancing this endogenous inhibition?

## 2. Materials and Methods

### 2.1. Subjects, Samples and Ethical Approval

To investigate the relationship between endogenous ACE inhibition and HSA concentration, 161 adults were enrolled from the staff of the Department of Cardiology, University of Debrecen, and from patients attending its outpatient clinic. None of the participants were receiving ACE inhibitor therapy, as verified by medication history and biochemical measurements [21]. All participants provided written informed consent prior to inclusion. The study protocol was approved by the Scientific and Research Ethics Committee of the University of Debrecen and by the Scientific and Research Ethics Committee of the Medical Research Council, Budapest, Hungary (ETT TUKEB, research authorization number: 33327-1/2015/EKU; approved on 23 June 2015). Basic information and key laboratory findings of this group are provided in the Appendix B Table A1.

For the analysis of the association between endogenous ACE inhibition and total FFA concentration, laboratory results from 101 serum samples submitted for diagnostic ACE activity measurement to the Division of Clinical Physiology, Department of Cardiology, University of Debrecen, were analyzed. This part of the study was approved by the Regional and Institutional Ethics Committee, Clinical Centre, University of Debrecen (research authorization number: 5925-2021, approved on 15 December 2021). Basic information and key laboratory findings of this group are provided in the Appendix B Table A2.

Blood samples were collected from all participants using standard aseptic technique into Vacutainer tubes (Cat. No. 367955, Becton Dickinson, Franklin Lakes, NJ, USA). Serum was separated after clotting by centrifugation at 1500× *g* for 15 min at room temperature. The serum samples were stored at −20 °C until analysis.

### 2.2. Production and Purification of Recombinant, Human ACE

The plasmid used for recombinant human ACE (recACE) production was obtained from GeneCopoeia (custom clone, N-terminal His-tagged; Cat. No. CS-Y4341-I03-02, Rockville, MD, USA). The plasmid sequence was codon-optimized for expression in insect cells, and the coding sequence corresponding to the membrane-anchoring region was deleted (see Appendix A. for the plasmid sequence). The plasmid (10 ng) was introduced into DH10Bac competent *E. coli* cells (100 µL, Cat. No. 10361-012, Thermo Fisher Scientific, Waltham, MA, USA) by heat shock (45 s, 42 °C) according to the manufacturer’s instructions. The resulting bacmids were isolated (HiSpeed Plasmid Maxi Kit, Cat. No. 12662, Qiagen, Hilden, Germany) and transfected into Sf9 insect cells (Cat. No. 11496-015, Thermo Fisher Scientific) using Cellfectin II reagent (Cat. No. 10362100, Thermo Fisher Scientific, Waltham, MA, USA; 480 ng bacmid in SF900 medium containing 40 µg/mL Cellfectin II). The produced baculoviruses were harvested and used to infect fresh Sf9 cultures. Insect cells were maintained in serum-free medium (Sf-900 II SFM, Cat. No. 10902153, Thermo Fisher Scientific, Waltham, MA, USA) supplemented with 1 µM ZnCl_2_ at 28 °C. Recombinant protein production was monitored by measuring ACE activity in the culture supernatant. recACE appeared in the medium and reached its maximum enzymatic activity on the fourth day post-infection. The supernatant containing recACE was collected, and the recombinant protein was purified by HPLC using Ni-NTA Superflow agarose (Cat. No. 25215, Thermo Fisher Scientific, Waltham, MA, USA). The supernatant containing recACE was collected, and the recombinant protein was purified by immobilized metal affinity chromatography on a Ni–NTA Superflow agarose column (Cat. No. 25215; Thermo Fisher Scientific, Waltham, MA, USA). Two milliliters of recACE-containing medium were loaded onto a column packed with 10 mL resin at a constant flow rate of 1 mL/min. The sample was applied over 10 min in equilibration buffer (200 mM Na_3_PO_4_, 300 mM NaCl, 10 mM imidazole; pH 7.4), followed by a 10-min wash step (20 mM Na_3_PO_4_, 300 mM NaCl, 20 mM imidazole; pH 7.4) and a 10-min elution step (20 mM Na_3_PO_4_, 300 mM NaCl, 300 mM imidazole; pH 7.4). Fractions eluting between 24 and 28 min were concentrated using 100 kDa MWCO centrifugal concentrators (Vivaspin 500; Cat. No. Z614092; Merck, Darmstadt, Germany), and the purified protein was stored in 100 mM Tris buffer (pH 7.0) containing 50% (*v*/*v*) glycerol at −20 °C until use.

### 2.3. Measurement of ACE Activity, Endogenous ACE Inhibition, HSA and Total FFA Concentration

ACE activity was determined using our optimized fluorescent kinetic assay, published elsewhere [22]. The level of serum HSA-mediated endogenous ACE inhibition was calculated from dilution-corrected ACE activity values measured at 4-fold and 400-fold serum dilutions, as described previously [21]. Serum HSA concentration was measured using a Cobas Integra 400 Plus clinical chemistry analyzer (Roche Diagnostics, Basel, Switzerland) with the manufacturer’s bromocresol green assay (Cat. No. 03183688122). Total FFA concentration was determined on the same analyzer by a colorimetric enzymatic method using the Dialab FFA reagent (Cat. No. D07940, Dialab, Neudorf, Austria), together with the corresponding calibrator (Diacal Lipids, Cat. No. D13585SV) and control materials (normal control: Cat. No. D99486; pathological control: Cat. No. D11487), according to the manufacturer’s instructions.

### 2.4. Removal of FFAs from HSA

HSA solution (Albunorm 200 g/L infusion, Octapharma, Lachen, Switzerland) was diluted to a final concentration of 100 g/L with distilled water. Activated charcoal powder (Cat. No. C9157-500G, Sigma-Aldrich, St. Louis, MO, USA) was added to the solution at a final concentration of 20% (*w*/*v*). The suspension was incubated for 24 h at room temperature with continuous rotation (15 rpm), followed by centrifugation at 20,000× *g* for 30 min at +4 °C to remove the charcoal. The resulting FFA-free HSA solution was concentrated to approximately 200 g/L using a miVac DUO Concentrator (GeneVac Inc., Ipswich, UK) and sterilized by filtration through a 0.2 μm membrane filter (Cat. No. 4652, Pall Life Sciences, Cornwall, UK). The HSA concentration was determined as described above.

### 2.5. Treatment of FFA-Free HSA with Specific Fatty Acids

The sodium salt of each fatty acid was mixed with FFA-free HSA at a 4:1 molar ratio. The mixture contained 50 mM NaCl and was incubated overnight at room temperature with continuous rotation (15 rpm). After incubation, unbound fatty acids were removed by filtration using a Vivaspin 20 concentrator (30 kDa cut-off; Cat. No. 28-9323-61, GE Healthcare, Buckinghamshire, UK). The treated HSA was then washed three times with distilled water, and after each washing step, the sample was reconcentrated using the same ultrafiltration device. HSA concentration was determined by the bromocresol green method, as described above. The procedure was repeated for each tested fatty acid. The physicochemical properties of the examined fatty acids are summarized in Appendix A.

### 2.6. Immunoprecipitation of Tissue ACE

High-binding 96-well ELISA plates (Cat. No. 655061, Greiner Bio-One, Kremsmünster, Austria) were coated with 0.6 μg/well goat anti-mouse IgG (cat. No. 31168, Pierce, Rockford, IL, USA) to immobilize the mouse monoclonal anti-ACE antibody 9B9 (a kind gift from Prof. Sergei Danilov) [23]. Plates were incubated for 2 h at 37 °C with continuous horizontal shaking (200 rpm), followed by washing with PBST (PBS (Cat. No. 14190-094, ThermoFisher, Waltham, MA USA) containing 0.05% Tween-20). The wells were then coated with 0.5 μg/well of the 9B9 antibody (prepared as 10 μg/mL in PBS containing 1% BSA Cat. No. A7906, Merck KGaA, Darmstadt, Germany) and washed twice with PBST before sample loading. Human tissue samples were mechanically pulverized in liquid nitrogen using a mortar and pestle. On ice, 5 mL of 100 mM Tris–HCl (pH 7.0) was added per gram of tissue (wet weight). The samples were homogenized using a tissue homogenizer (Bio-Gen PRO200; PRO Scientific, Oxford, CT, USA) and centrifuged at 16,100× *g* for 5 min. The supernatants were collected and diluted 5–20-fold in PBS containing 1% (*w*/*v*) BSA and added to the antibody-coated wells. Plates were incubated under the same conditions as above, then washed twice with PBST and once with PBS. The prepared plates were subsequently used for HSA treatments. Serial twofold dilutions of control HSA and FFA-free HSA, starting from a final concentration of 90 g/L, were added to the wells, and ACE activity was measured as described in Section 2.3.

### 2.7. Statistical Analysis

Dose–response data were fitted using a four-parameter logistic inhibition model. IC_50_ values and 95% confidence intervals (profile likelihood) were obtained from the fits. Differences between IC_50_ values were evaluated using the extra-sum-of-squares F-test (‘Compare fits’ option). To account for multiple testing across planned comparisons of FFA-loaded HSA versus fatty acid–free HSA, *p*-values were adjusted using the Holm–Šidák method. Multivariable linear regression (ordinary least squares) was used to assess the independent associations of serum HSA concentration and other covariates with endogenous ACE inhibition. Models including additional clinical covariates were performed as complete-case analyses. Model diagnostics included assessment of residual normality and multicollinearity (variance inflation factors, VIF). Continuous variables were tested for normality and are presented as median [interquartile range]; group differences were assessed using the Mann–Whitney U test. Categorical variables were compared using the chi-square (χ^2^). A *p* value < 0.05 was considered statistically significant. Statistical analyses were performed using GraphPad Prism v10 (GraphPad Software, San Diego, CA, USA).

### 2.8. Role of Generative AI and AI-Assisted Technologies in the Writing Process

During the preparation of this work, the authors used ChatGPT (OpenAI, San Francisco, CA, USA, version 5.2) to improve the clarity, structure, and readability of the manuscript, including the abstract and cover letter. After using this tool, the authors carefully reviewed and edited the content and take full responsibility for the integrity and accuracy of the publication.

## 3. Results

### 3.1. Endogenous ACE Inhibition Depends on HSA Concentration and Its Bound Non-Aqueous Compounds

Endogenous ACE inhibition and HSA concentration were measured in 161 individuals (Figure 1a). The degree of inhibition increased with higher HSA levels (Spearman’s rho = 0.422, 95% CI 0.281–0.544, *p* < 0.0001). However, the linear fit to the data was only moderate (r^2^ = 0.164), suggesting that—besides HSA concentration—other factors may also influence the extent of endogenous ACE inhibition. To further explore potential confounders, we performed multivariable linear regression analyses. HSA concentration remained positively associated with endogenous ACE inhibition after adjustment for age and sex (Table A3), and this finding was confirmed in complete-case and extended models including BMI and eGFR (Table A4 and Table A5). Analyses in the smaller bilirubin subset are presented as exploratory (Table A6).

HSA serves as a major carrier protein for both water-soluble and water-insoluble molecules. Therapeutic HSA preparations are typically purified by dialysis, which effectively removes water-soluble substances but leaves water-insoluble compounds bound to HSA. HSA purified by dialysis markedly reduced recombinant ACE activity in a concentration-dependent manner (IC_50_ = 7.84 [6.58–9.75] g/L), resulting in approximately 90% inhibition at physiological HSA concentration (at 45 g/L; Figure 1b, blue squares). In contrast, removal of water-insoluble compounds from HSA by active-charcoal treatment significantly diminished its ACE-inhibitory capacity (IC_50_ = 23.24 [19.40–29.78] g/L, *p* < 0.001; Holm–Šidák-adjusted *p* = 0.001; Figure 1b, red circles), leading to a roughly fourfold increase in residual ACE activity at physiological HSA levels (41% at 45 g/L HSA concentration).

### 3.2. The Type, Rather than the Total Concentration of FFAs Determines the Extent of Endogenous ACE Inhibition

Measurement of serum FFA concentration and its comparison with the degree of endogenous ACE inhibition revealed no significant association between these parameters (Figure 2a; Spearman’s rho = 0.088, *p* = 0.38, r^2^ = 0.008, n = 101). This suggests that the total FFA concentration is not a key determinant of endogenous ACE inhibition. We hypothesized that the lack of correlation may arise from the fact that different FFA species exert distinct effects on HSA-mediated ACE inhibition. This lack of association remained unchanged in multivariable linear regression adjusting for age, sex, and serum HSA (Appendix B Table A7), and in sensitivity models additionally including hsCRP and eGFR (Appendix B Table A8) or total bilirubin as a HSA ligand (Appendix B Table A9).

To test this hypothesis, fatty acid–free HSA was treated with various saturated FFAs (Figure 2b). Treatment of HSA with dodecanoic acid (C12:0) did not significantly alter its ACE-inhibitory capacity compared to fatty acid–free HSA (IC_50_ = 18.69 [15.36–24.97] g/L; inhibition at 45 g/L HSA = 67.9 ± 0.4% vs. FFA-free HSA IC_50_ = 23.24 [19.40–29.78] g/L; inhibition at 45 g/L HSA = 59.4 ± 1.4%; *p* = 0.17). In contrast, treatment with myristic acid (C14:0) or stearic acid (C18:0) markedly enhanced the ACE-inhibitory effect of HSA (C14:0: IC_50_ = 13.26 [10.19–19.86] g/L; inhibition at 45 g/L HSA = 77.1 ± 1.0%; *p* = 0.006; Holm–Šidák-adjusted *p* = 0.023; C18:0: IC_50_ = 7.98 [7.04–9.23] g/L; inhibition at 45 g/L HSA = 87.4 ± 0.6%; *p* < 0.001, Holm–Šidák-adjusted *p* = 0.001).

### 3.3. The Chain Length of Saturated Fatty Acids Determines the Magnitude of HSA-Mediated ACE Inhibition

Our previous findings suggested that the chain length of FFAs may influence the extent of HSA-mediated ACE inhibition. To further investigate this relationship, concentration–response curves were established for saturated FFAs with different carbon chain lengths (Appendix A). Based on these data, we determined the level of ACE inhibition at physiological HSA concentration (Figure 3a) and calculated the corresponding IC_50_ values (Figure 3b). Among the tested FFAs, stearic acid (C18:0) exerted the strongest HSA-mediated inhibitory effect on ACE activity (87.4 ± 0.6%, *p* < 0.001, Holm–Šidák-adjusted *p* = 0.002), while capric acid (C10:0) showed no appreciable effect compared to fatty acid–free HSA (59.0 ± 0.5%, *p* = 0.468, Holm–Šidák-adjusted *p* = 0.932). In parallel, the IC_50_ values of the FFA–HSA complexes gradually decreased from capric acid (C10:0, IC_50_ = 27.09 [19.39–55.55] g/L) to stearic acid (C18:0, IC_50_ = 7.98 [7.04–9.23] g/L), followed by a slight increase with longer-chain FFAs (Figure 3b).

### 3.4. The Presence and Position of a Double Bond Modulate the HSA-Mediated ACE-Inhibitory Effect

We examined whether the presence of a double bond in fatty acids of identical chain length affects the HSA-mediated inhibition of ACE (Figure 4). Elaidic acid, which contains a trans double bond at the n-7 position compared to stearic acid, retains a linear structure despite being unsaturated. This structural modification markedly altered its modulatory effect on the ACE-inhibitory capacity of HSA (C18:1 (n-7t), IC_50_ = 10.74 [9.71–12.05] g/L; inhibition at 45 g/L HSA = 80.7 ± 0.7%, vs. stearic acid C18:0, IC_50_ = 7.98 [7.04–9.23] g/L; inhibition at 45 g/L HSA = 87.4 ± 0.6%, *p* = 0.001, Figure 4a). In contrast, cis-vaccenic acid, which also contains a double bond at the n-7 position but in the cis configuration, did not show a significant difference in HSA-mediated ACE inhibition compared with stearic acid (C18:1 (n-7), IC_50_ = 8.70 [7.85–9.79] g/L; inhibition at 45 g/L HSA = 84.3 ± 0.7%, *p* = 0.317, Figure 4b). However, when this cis double bond was located at the n-9 position, as in oleic acid, the enhancement of HSA-mediated ACE inhibition was significantly lower (C18:1 (n-9), IC_50_ = 11.61 [10.35–13.28] g/L; inhibition at 45 g/L HSA = 80.1 ± 1.7%, *p* < 0.001).

### 3.5. Certain Omega-3 and Omega-6 Fatty Acids Markedly Enhance the HSA-Mediated ACE-Inhibitory Effect

We next investigated whether polyunsaturated omega-3 and omega-6 fatty acids modulate the HSA-induced endogenous ACE inhibition. Among the omega-3 fatty acids tested (Figure 5a), α-linolenic acid produced the most pronounced enhancement of HSA mediated ACE inhibition (C18:3 (n-3), IC_50_ = 5.60 [4.28–6.15] g/L; inhibition at 45 g/L HSA = 89.0 ± 0.3%, *p* < 0.001, Holm–Šidák-adjusted *p* = 0.001), followed by docosahexaenoic acid (DHA), which showed a somewhat weaker yet still significant effect compared with FFA-free HSA (C22:6 (n-3), IC_50_ = 12.02 [9.86–15.06] g/L; inhibition at 45 g/L HSA = 82.6 ± 0.9%, *p* < 0.001, Holm–Šidák-adjusted *p* = 0.001). In contrast, the presence of eicosapentaenoic acid (EPA) on HSA did not significantly modify its IC_50_ value (C20:5 (n-3), IC_50_ = 16.56 [13.00–24.22] g/L; inhibition at 45 g/L HSA = 78.9 ± 0.9%, *p* = 0.059, Holm–Šidák-adjusted *p* = 0.114). Regarding omega-6 fatty acids (Figure 5b), γ-linolenic acid exhibited the strongest potentiating effect on the HSA-mediated ACE inhibition (C18:3 (n-6), IC_50_ = 5.09 [4.28–6.15] g/L; inhibition at 45 g/L HSA = 91.7 ± 0.2%, *p* < 0.001, Holm–Šidák-adjusted *p* = 0.001), while eicosadienoic acid and linoleic acid also significantly enhanced the inhibitory capacity, although to a lesser extent (C20:2 (n-6), IC_50_ = 14.41 [11.79–19.06] g/L; inhibition at 45 g/L HSA = 76.1 ± 1.7%, *p* = 0.001, Holm–Šidák-adjusted *p* = 0.023; C18:2 (n-6), IC_50_ = 15.04 [13.55–17.00] g/L; inhibition at 45 g/L HSA = 72.8 ± 1.3%, *p* = 0.001, Holm–Šidák-adjusted *p* = 0.005).

### 3.6. The HSA-Mediated ACE-Inhibitory Effect of FFAs Is Also Observed in Tissue-Derived ACE

To assess whether the effect of FFAs on HSA-mediated ACE inhibition can also be detectable in tissue-derived ACE, we examined lung, lymph node, and cardiac tissue homogenates (Figure 6). ACE was immobilized onto ELISA plate surfaces using anti-ACE antibodies, followed by washing to remove unbound tissue proteins. When the immobilized ACE was exposed to HSA containing physiologically bound FFAs, a markedly stronger inhibition was observed compared to reactions containing fatty acid–free HSA. This difference was consistent across all tested tissues, including lung (IC_50_ = 5.52 [4.07–7.90] g/L; inhibition at 45 g/L HSA = 88.6 ± 2.1%, vs. IC_50_ = 19.55 [14.55–32.86] g/L; inhibition at 45 g/L HSA = 66.7 ± 5.2%, *p* < 0.001; Figure 6a), lymph node (IC_50_ = 5.52 [4.08–8.35] g/L; inhibition at 45 g/L HSA = 86.0 ± 1.6%, vs. IC_50_ = 13.40 [7.99–128.90] g/L; inhibition at 45 g/L HSA = 71.7 ± 1.2%, *p* = 0.019; Figure 6b), and cardiac homogenates (IC_50_ = 1.69 [1.36–2.03] g/L; inhibition at 45 g/L HSA = 84.7 ± 1.2%, vs. IC_50_ = 7.59 [5.01–17.38] g/L; inhibition at 45 g/L HSA = 54.41 ± 3.2%, *p* < 0.001; Figure 6c).

## 4. Discussion

Cardiovascular diseases remain the leading cause of death globally, accounting for an estimated 19.8 million deaths in 2022—about one in three deaths worldwide [24]. Therapeutic inhibition of ACE with ACE inhibitor drugs has been shown to reduce cardiovascular mortality by approximately 20–25% compared to other antihypertensive treatments [25], highlighting the essential role of ACE in cardiovascular pathology. We have previously reported that human serum albumin (HSA) endogenously inhibits circulating ACE activity [9,26], and that the lack of this inhibition may contribute to the development of cardiovascular diseases [12]. This hypothesis is further supported by numerous studies demonstrating that low-normal serum HSA concentration is an independent risk factor for cardiovascular morbidity and mortality [11,27,28].

In the present study, we demonstrate that the level of endogenous ACE inhibition is linearly related to HSA concentration. Higher circulating HSA levels may reduce the formation of the vasoconstrictor angiotensin II through ACE inhibition, suggesting that HSA potentially contribute to blood pressure reduction. Consistent with this hypothesis, other research groups have observed that HSA infusion can induce hypotension [29,30], and that high-normal serum HSA levels may protect against the development of hypertension [31,32].

HSA is a major carrier protein in the bloodstream, capable of binding both hydrophilic and hydrophobic molecules as well as a wide variety of drugs [13]. Our present results indicate that albumin-bound FFAs are key modulators of the HSA-mediated endogenous inhibition of ACE, as the removal of FFAs from HSA markedly diminished its inhibitory capacity. HSA possesses multiple well-characterized fatty acid binding sites with high or moderate affinity, and FFA binding to these sites induces substantial conformational changes in the protein structure [13,14,33]. These conformational changes may influence HSA–ACE interactions and thereby modulate the ACE-inhibitory efficacy of HSA. This concept is consistent with observations showing that the tyrosine residue (Tyr-138) in the FA1 site in subdomain IB plays a critical role in both FFA binding [33] and in the development of endogenous ACE inhibition [34]. Moreover, another potential ACE-binding site of HSA (Ala-213–Arg-218) [35] co-localizes with Sudlow site I (drug site I, FA7) in subdomain IIA [36], providing a plausible additional structural basis for the modulatory effect of FFAs.

Although elevated total FFA concentration in the circulation has been associated with insulin resistance [37] and type 2 diabetes mellitus [38], as well as with increased cardiovascular risk [39,40], growing evidence indicates that the qualitative composition of FFAs rather than their total amount determines their physiological impact [41,42,43]. In this context, our data show that total FFA concentration was not associated with endogenous ACE inhibition in our cohort, whereas our in vitro experiments indicate that the type of fatty acid bound to HSA can markedly alter this inhibitory capacity. Thus, beyond its transport role, HSA may also act as a regulator of ACE activity, with its inhibitory potency influenced by the composition of bound fatty acids.

In our experiments, among saturated fatty acids, stearate and palmitate most effectively enhanced HSA-mediated endogenous ACE inhibition. This observation is particularly relevant given that in obesity and insulin-resistant states, circulating FFA levels—especially palmitate and, to a lesser extent, stearate—are frequently elevated [44,45]. These findings raise the possibility that, under such metabolic conditions, shifts in the composition of HSA-bound fatty acids may enhance endogenous ACE inhibition and contribute to partial compensation against RAAS overactivation; however, the in vivo relevance of this concept remains to be determined.

Regular physical activity is a cornerstone of cardiovascular prevention and has been consistently associated with improved metabolic flexibility and reduced cardiovascular risk. Exercise enhances whole-body lipolysis and increases the release of FFAs such as palmitate and, in some settings, stearate, both at rest and during exertion [45,46]. Consequently, the elevated availability of HSA-bound FFAs could transiently amplify endogenous ACE inhibition, which may represent one potential mechanism contributing to the well-established cardioprotective effects of regular physical activity through modulation of the renin–angiotensin system.

Contemporary guidelines and evidence syntheses draw a nuanced picture regarding omega-3 fatty acids in cardiovascular prevention. Broad use of long-chain EPA/DHA supplements does not meaningfully reduce cardiovascular events or mortality [47]. Current practice guidelines restrict therapeutic recommendations largely to purified EPA in selected high-risk patients with persistently elevated triglycerides on statins and do not recommend routine fish-oil supplementation for general prevention [48]. In this context, we show that specific polyunsaturated fatty acids—most notably α-linolenic acid (ALA, 18:3 n-3) and γ-linolenic acid (GLA, 18:3 n-6)—markedly enhance HSA-mediated endogenous ACE inhibition, exceeding the effect of saturated fatty acids; at physiological HSA concentrations, ALA-bound HSA yielded ~90% inhibition and GLA-bound HSA 91.7% in our assays. These observations are compatible with human evidence: in a double-blind, randomized trial in patients with peripheral artery disease, daily ingestion of 30 g milled flaxseed (rich in ALA) for 6 months lowered systolic and diastolic blood pressure by ~10/7 mmHg versus placebo, with even larger reductions in those hypertensive at baseline (−15/−7 mmHg); circulating ALA levels rose in parallel [17]. Beyond single trials, recent meta-analyses of randomized controlled trials confirm that flaxseed interventions produce significant blood-pressure reductions, with larger effects in hypertensive cohorts [49]. A systematic review further reported that higher dietary ALA intake is associated with modestly lower risks of cardiovascular and coronary mortality [16]. One possible mechanism is that HSA-bound ALA/GLA may augment HSA-mediated ACE inhibition and thereby attenuate Ang II generation; however, this link to blood pressure lowering was not directly tested in the present study.

Our tissue-ACE–based data indicate that FFA-modulated, HSA-mediated regulation of ACE is not confined to the vascular lumen but extends to tissue ACE pools (myocardium, lung, lymphoid tissues). From a cardiovascular standpoint, ACE within the pulmonary vasculature and myocardium is central to local angiotensin II generation and hemodynamic control. Equally, ACE is expressed in immune cells and lymphoid organs, where it contributes to antigen processing and inflammatory tone [50]. Accordingly, qualitative shifts in the HSA-bound fatty-acid cargo (e.g., ALA/GLA enrichment or metabolic/inflammatory remodeling) could not only influence vascular RAAS activity but also fine-tune tissue and immune ACE activity within interstitial and lymphatic compartments via HSA-mediated endogenous inhibition. Together, these observations raise the hypothesis that diet- or disease-driven changes in HSA-bound fatty acids could influence both cardiovascular and immune-related ACE activity; this broader concept remains hypothesis-generating and warrants dedicated investigation.

Limitations: This study has limitations: (1) although activated charcoal–based defatting is a widely used approach to generate fatty acid–free HSA, it may also remove other hydrophobic HSA-bound ligands and/or subtly affect HSA conformation/aggregation, which could influence HSA-ACE interactions beyond FFA removal; (2) our in vitro reconstitution used a fixed FFA:HSA molar ratio (4:1) to ensure reproducible loading and to capture the maximal inhibitory potential of each fatty acid species, which may not fully reflect typical physiological FFA occupancy; (3) despite multivariable analyses, residual confounding cannot be excluded, as endogenous ACE inhibition may be influenced by unmeasured factors (e.g., detailed lipid profile parameters); and (4) because individual FFA species were not quantified in patient samples, clinical associations are limited to total FFA/FFA:HSA ratio and cannot be attributed to specific circulating fatty acids in vivo, which would require targeted lipidomics/FFA speciation.

## 5. Conclusions

Our data support that endogenous ACE inhibition is associated with serum HSA concentration and can be modulated by the qualitative composition of HSA-bound fatty acids. In vitro reconstitution experiments indicate that several FFAs—particularly long-chain saturated species and polyunsaturated ALA/GLA—enhance the ACE-inhibitory capacity of HSA, with effects observed not only on recombinant ACE but also on tissue-associated ACE. Collectively, these findings position albumin-bound FFAs as potentially important modulators of ACE/RAAS regulation and suggest that changes in metabolic milieu and fatty acid composition may be relevant to cardiovascular physiology.

## Figures and Tables

**Figure 1 biomedicines-14-00103-f001:**
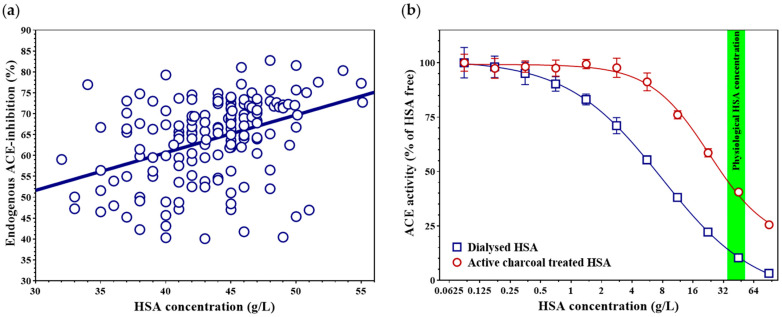
Dependence of endogenous ACE inhibition on HSA concentration (**a**). Blue circles represent the individual data points of 161 subjects. The thick blue line indicates the best-fit regression line. Effect of dialyzed (blue squares) and charcoal-treated (red circles) HSA on recombinant ACE activity (**b**). Symbols represent the mean and standard deviation of three independent measurements. The physiological HSA concentration range (35–52 g/L) is indicated by a green band.

**Figure 2 biomedicines-14-00103-f002:**
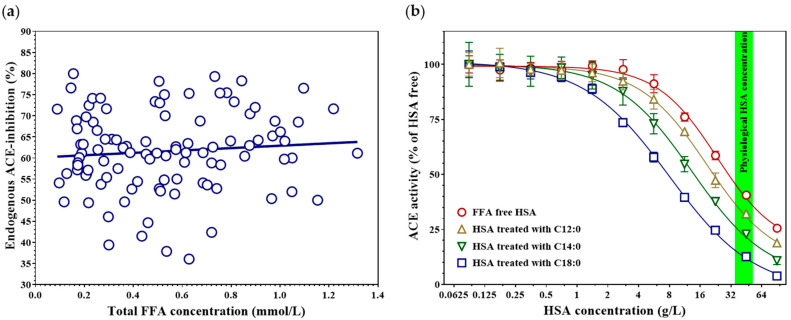
Serum FFA concentration plotted against the degree of endogenous ACE inhibition (**a**). Blue circles represent individual data points of 101 subjects. The blue line indicates the best-fit regression line. Concentration–response curves showing the effect of different saturated FFAs bound to fatty acid–free HSA on recombinant ACE activity (**b**). Dodecanoic acid (C12:0, mustard-yellow triangles), myristic acid (C14:0, green inverted triangles), and stearic acid (C18:0, blue squares) were added to HSA, and the resulting complexes were tested for their ACE-inhibitory capacity. The effect of FFA-free (control) HSA is indicated by red circles. Symbols represent the mean ± SD of three independent measurements. The physiological serum HSA concentration range (35–52 g/L) is indicated by the green band.

**Figure 3 biomedicines-14-00103-f003:**
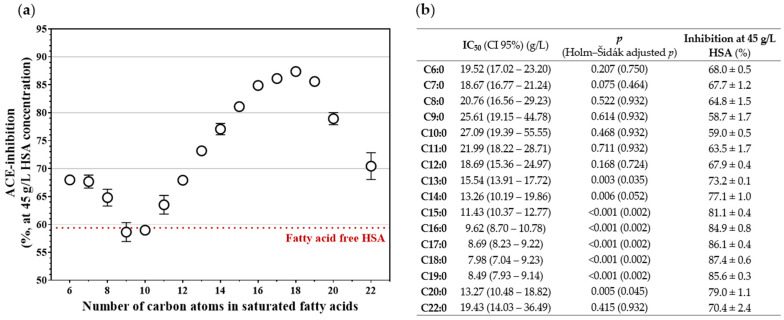
ACE inhibition measured at physiological HSA concentration (45 g/L) as a function of the number of carbon atoms in saturated fatty acids (**a**). Circles represent the mean ± SD of three independent measurements. The dotted red line indicates the ACE inhibition level obtained with fatty acid–free HSA. IC_50_ values (95% confidence intervals) and the corresponding ACE inhibition at 45 g/L HSA (mean ± SD) are shown for saturated fatty acids ranging from C6:0 to C22:0 in panel (**b**), expressed as a percentage of HSA-free ACE activity, with raw and Holm–Šidák-adjusted *p* values reported for comparisons versus fatty acid–free HSA.

**Figure 4 biomedicines-14-00103-f004:**
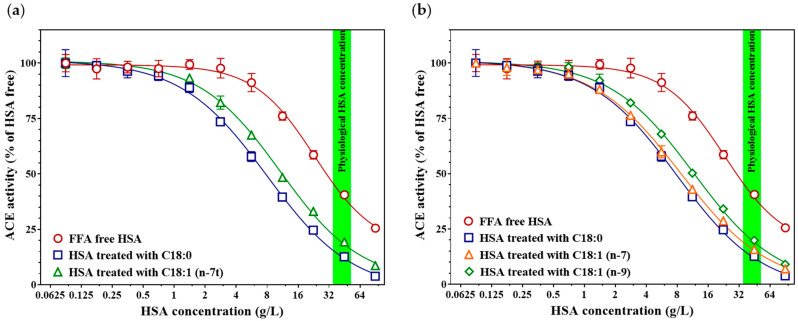
Effects of elaidic acid (C18:1 (n-7t), green triangles) and stearic acid (C18:0, blue squares) on HSA-mediated ACE inhibition (**a**). The concentration–response curve of FFA-free (control) HSA on recombinant ACE activity is shown in red. Effects of cis-vaccenic acid (C18:1 n-7 cis, orange triangles) and oleic acid (C18:1 n-9 cis, green diamonds) on HSA-mediated ACE inhibition (**b**). Stearic acid (C18:0, blue squares) and fatty acid–free HSA control (red circles) are also included for comparison. The green band indicates the physiological serum HSA concentration range (35–52 g/L). Symbols represent the mean ± SD of three independent measurements.

**Figure 5 biomedicines-14-00103-f005:**
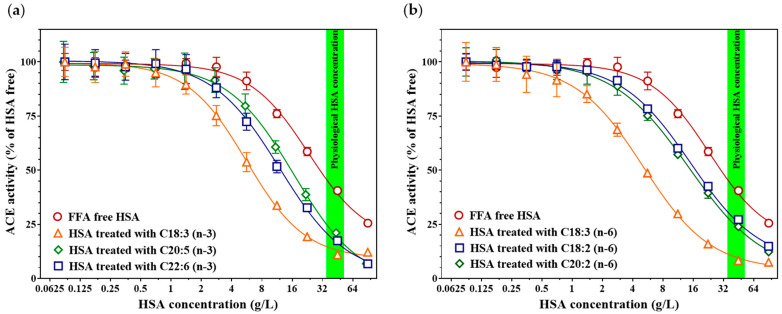
Effects of α-linolenic acid (C18:3 (n-3), orange triangles), docosahexaenoic acid (C22:6 (n-3), blue squares) and eicosapentaenoic acid (C20:5 (n-3), green diamonds) on HSA-mediated ACE inhibition (**a**). The concentration–response curve of FFA-free (control) HSA on recombinant ACE activity is shown in red. Effects of γ-linolenic acid (C18:3 n-6, orange triangles), eicosadienoic acid (C20:2 n-6, green diamonds), and linoleic acid (C18:2 n-6, blue squares) on HSA-mediated ACE inhibition (**b**). Fatty acid–free HSA control (red circles) is also included for comparison. The green band indicates the physiological serum HSA concentration range (35–52 g/L). Symbols represent the mean ± SD of three independent measurements.

**Figure 6 biomedicines-14-00103-f006:**
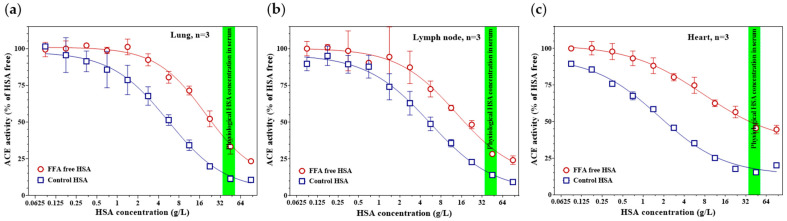
The ACE-inhibitory effects of HSA containing physiologically bound FFAs (blue squares) and fatty acid–free HSA (red circles) were tested using ACE immunoprecipitated from lung (**a**), lymph node (**b**), and cardiac (**c**) tissue homogenates. Representative graphs show the mean ± SD of three independent measurements. The green band indicates the physiological serum HSA concentration range.

## Data Availability

The raw data can be obtained on request from the corresponding author.

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
