# Peer review of "Albumin-Bound Fatty Acids Modulate Endogenous Angiotensin-Converting Enzyme (ACE) Inhibition"

_biomedicines, 2026, doi:10.3390/biomedicines14010103_

Round 1

Reviewer 1 Report

Comments and Suggestions for Authors

The manuscript entitled " Albumin-bound fatty acids modulate endogenous angiotensin-2 converting enzyme (ACE) inhibition" contains interesting study. The topic is interesting and the data are unique but the interpretation needs refining.

  • The rational for this study is still questionable. There are many drugs with good tolerability in this category or ARB.
  • The authors should confirm the albumin-FFA conjugation with an appropriate characteristics test.
  • How did the authors ensure that just free FFA adsorb to the charcoal?
  • The authors should compare the IC50 with each other with an appropriate statistical analysis test and interpret the results.
  • The authors should unify the style of all references according to the authors guideline.
  • It is recommended to compare the results in this study with a drug in this category (e.g., Captopril, Enalapril) used in clinic.

Reviewer 2 Report

Comments and Suggestions for Authors

I congratulate the authors for this novel and potentially important study regarding the  importance of albumin-bound fatty accids in modulating endogenous ACE inhibition. Bwloe are some comments that should be addressed before the study may be considered for publishing:

 - line 93 - is nationality/citizenship relevant? If no, I suggest removing "Hungarians"

  • the authors should add other baseline characteristics, which might influence both albumin and ACE, and which should further be used for multivariate analysis, such as BMI, lipid profile, inflammatory markers, and kidney function markers:
  • the exclusion criteria focus only on ACE inhibitor therapy; however, there are other RAAS-modulating drugs, such as ARBs or aldosterone agonists, which are not mentioned. If they were not excluded, the validity of this study is significantly put into question.
  • there are no direct measurements of individual FFAs; the authors only measure total FFA concentration in patient samples, and then jump to conclusions about specific fatty acids based solely on in vitro experiments with purified, artificially loaded HSA;
  • typical in vivo FFA:HSA molar ratios in healthy fasting adults are roughly in the range of about 0.1–1:1, and ratios above ~2–3:1 are generally considered elevated or supraphysiologic; therefore, the 4:1 molar ratio of FFA:HSA used in experiments (Section 2.5) far exceeds physiological ratios and may not properly reflect in vivo conditions.
  • regarding charcoal use: while it removes water-insoluble compounds, activated charcoal may also alter HSA structure or remove essential cofactors beyond FFAs
  • the r2 values (fig. 1a, 2a) show a very weak correlation. The authors properly acknowledge this, but fail to adequately explore other contributing factors, which may be done through multivariate analysis (controlling for age, sex, comorbidities other albumin ligands; see also the previous comments)
  • the authors have employed multiple comparisons - testing numerous fatty acids without correcting for multiple tests signficantly increases the Type I error risk; this, associated with the fact that the difference between some fatty acids approaches non-significance puts into question the validity of the study;
  • I also have some reproducibility concerns - there are insuficient details about expression of ACE, protein purity, activity validation agains native ACE; regarding tissue preparation - there is insuficient data about homogenization protocols, storage conditions, ACE isoform validation.
  • the abstract overstates the conclusions - the use of constructs such as "critical determinant" or "suggests a link" seem to strong given the correlation nature and the emphasis on in vitro experimentation.

Reviewer 3 Report

Comments and Suggestions for Authors

I really liked this manuscript and as somebody who has worked in a related area for over 2 decades I was fascinated to learn about the properties of HSA on ACE1 function which I was not familiar with. I have just no major suggestions for the authors.

Round 2

Reviewer 1 Report

Comments and Suggestions for Authors

regards for improving the articoe. I would like to suggest its accept

Reviewer 2 Report

Comments and Suggestions for Authors

the authors have improved the article significantly. I have no further issues that should be corrected